# A Tight Coupling Algorithm for Strapdown Inertial Navigation System (SINS)/Global Positioning System (GPS) Adaptive Integrated Navigation Based on Variational Bayesian

**Jiaxin Liu** [1] , **Ke Di** [1], **Hui Peng** [1,2] **and Yu Liu** [1,2,*]

1. Chongqing Key Laboratory of Autonomous Navigation and Microsystems,
   Chongqing University of Posts and Telecommunications, Chongqing 400065, China;
   s190401011@stu.cqupt.edu.cn (J.L.); dike@cqupt.edu.cn (K.D.); penghui@cqupt.edu.cn (H.P.)
2. Chongqing Engineering Research Center of Intelligent Sensing Technology and Microsystem,
   Chongqing University of Posts and Telecommunications, Chongqing 400065, China
* Correspondence: liuyu@cqupt.edu.cn; Tel.: +86-138-8315-8002

**Abstract:** Multi-source nonlinear noise exists in the process of multi-source navigation information fusion in long-endurance positioning systems in complex environments. In such engineering applications, the classical Kalman filter (KF) and the extended Kalman filter (EKF) have the phenomena of noise instability and parameter drift, which lead to the divergence of filtering results and reductions in accuracy over long periods of time. Aiming at the above problems, this paper proposes a fusion algorithm of the variational Bayesian (VB) and the cubature Kalman filter (CKF). Firstly, the system is modeled through nonlinear filtering, and the CKF error equation is established by taking the position difference and velocity difference between SINS and GPS as observation variables. Then, to address the problem of poor self-adaptation of the CKF algorithm, the variational Bayesian adaptive estimation method is introduced into the CKF algorithm, and a measurement noise variance estimation model is introduced to the process of time and measurement updates of the CKF algorithm to finally obtain the adaptive VB–CKF algorithm. The simulation results from the experimental platform show that the proposed fusion algorithm improves the combined SINS/GPS system by about 30% in terms of attitude angle accuracy and reduces speed and position estimation errors (RMSE) by about 45%. At the same time, comprehensive experiments on multiple types of sites show that compared with the CKF algorithm, the VB–CKF algorithm improves the positioning accuracy by 10% when the GPS signal is stable and improves the accuracy by about 38% when the GPS measurement noise changes dramatically in complex terrain, which effectively suppresses the accuracy divergence of the CKF algorithm and has high value for engineering applications.

**Keywords:** SINS/GPS integrated system; nonlinear noise; adaptivity; VB–CKF algorithm

## 1. Introduction

Multi-source information fusion is an advanced positioning method that pinpoints a carrier's exact location by combining information from several navigational sources. It has been extensively used in both civil and military settings, including geodetic surveying and mining. Military settings include the sea, land, and air. A range of navigation methods, such as inertial navigation based on inertial devices and satellite navigation based on satellite location, are starting to emerge with advancements in science and technology. In practical applications for integrated positioning, two or more separate navigation methods are frequently combined to increase positioning accuracy [1]. Fusions of navigation and positioning technology combine the advantages of several navigation methods together. Among them, navigation systems combining the strapdown inertial navigation system (SINS), which uses inertial devices for calculating navigation, and the global navigation satellite system (GNSS) are mainstream.

The SINS collects data using inertial devices and then solves the position information by using a navigation algorithm. Since the location information it solves has strong anonymity and concealment, it is not disturbed, even in a complicated confined context. However, based on the navigational theory of the SINS and the inertial sensors' own low accuracy, the system's navigational error rises over time [2]. GNSS is a space-based wireless navigation system that provides navigation information by satellite, and it can precisely determine a receiver's position through the time difference between the signals received on the ground and the signals sent from the satellites. Because the GNSS positioning technique requires the real-time intervention of satellite signals during the whole operation, there is no cumulative positioning error caused by increases in working time. But, it also creates a problem, which is outlined as follows: since GNSS needs to receive signals in real time during the working process and because the quality of the received signals is related to the external environment, its positioning accuracy is easily affected by the external environment. Considering the coverage of ground monitoring stations and the construction and cost of use in the civil field, this paper uses the Global Positioning System (GPS), which is one of the branches of GNSS.

So, it can be found from the above that GPS can effectively address the impact of cumulative errors, while the SINS can compensate for the lack of positioning accuracy when GPS signals are disturbed. Combining two navigation and positioning technologies can increase the precision of navigation and positioning through the technologies enhancing each other's strengths [3]. The data from the two navigation sources can be fused by the proposed SINS/GPS integrated navigation system, and the fusion approaches can be split into loose coupling and tight coupling, depending on complementarity and scene requirements. The integrated navigation system's loose coupling is a straightforward design that uses GPS navigation data to repair mistakes made by SINS. GPS and SINS operate separately and without interfering with one another. With long-term work, this cascade structure results in issues of rapid error divergence and weak anti-interference ability [4].

The tight coupling structure is described as SINS and GPS correcting each other; the structure can take the difference between the pseudorange and pseudorange rate information obtained by GPS and the pseudorange and pseudorange rate information calculated by SINS combined with an ephemeris and then use this difference as the input of the fusion filter to correct estimation errors. Although this connection structure has a high computational cost, it performs long-endurance tasks with greater accuracy and robustness [5]. In the tight coupling system, the velocity and position solutions of SINS and GPS have nonlinear characteristics. The Kalman filter (KF) is a linear filtering technology that belongs to a special case of Bayesian filtering inference. Due to the existence of many unknown noises in practical applications, the classical linear Kalman filter has been unable to meet practical engineering needs [6]. Filters based on nonlinear filtering technology can help solve the problem of information fusion in a nonlinear state, so many nonlinear filtering algorithm ideas are derived from Bayesian filtering, and scholars in the industry have carried out a series of studies on this and derived a series of filtering methods. One of the more commonly used methods in combined systems is the extended Kalman filter (EKF), which sets the system to a nonlinear state and truncates the higher-order terms in the Taylor expansion. Because the noise in the system satisfies the properties of Gaussian distribution, the EKF is relatively simple to implement and converges very quickly [7]. However, the disadvantage of the EKF is that if the system is strongly linear or has large errors at higher orders, the estimation accuracy of the EKF deviates greatly, resulting in filter scattering and a lower accuracy after fusion. J. L. Carssidis [8] proposed a new SINS/GPS combination method based on the unscented Kalman filter (UKF) to ensure that the EKF is continuously differentiable in the system model, using the Rodriguez attitude update algorithm to avoid the effects of quadratic normalization. However, this method still has the phenomena of noise instability and parameter drift. H. U. Heo proposed a UKF adaptive filtering algorithm based on Interacting Multiple Model (IMM) fusion, which uses

a Gaussian density function to compute the nonlinear posterior distribution of the system state equation [9]. The experimental results show that this fusion algorithm is better than the filtering effect of EKF and UKF alone; even without knowing the accurate state model and noise variance of the SINS/GPS integrated system, the fusion result is ideal and more adaptive, although the structure of its multi-cascade filter makes its implementation very complicated. In addition, the complexity of the UKF algorithm increases rapidly with an increase in the state and observation dimension, which eventually leads to poor real-time performance [10].

The complexity of the cubature transformation Kalman filter (CKF), in contrast, is considerably less than that of the UKF because it uses the cubature numerical integration concept and solely considers the state dimension when determining transition weights. As a result, to address the aforementioned issues that the SINS/GPS integrated navigation system frequently encounters, this article uses the CKF to fuse the SINS and GPS signals, as illustrated in Figure 1, in order to improve the stability and accuracy of long-endurance navigation. Additionally, in order to improve the adaptability of the CKF algorithm, it is necessary to choose an appropriate estimator to account for the noise statistical characteristics of the combined system given the estimation error brought on by the challenge of determining various time-varying noise models in practical engineering applications [11]. Finally, the superiority of the positioning and navigation performance of the algorithm is verified by experiments.

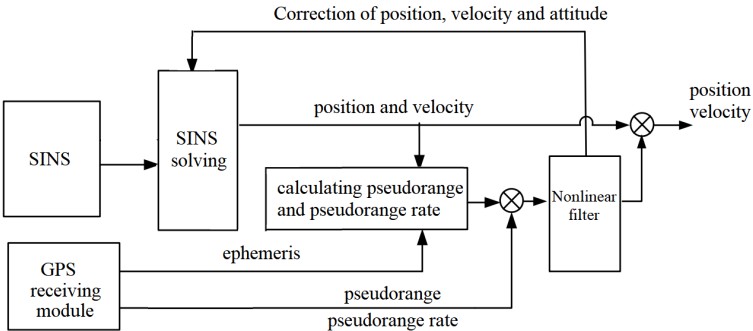

**Figure 1.** Schematic diagram of the SINS and GPS fusion system using CKF.

The SINS/GPS integrated navigation system has two types of error feedback: open-loop feedback and closed-loop feedback. The primary role of the combined system's filter in closed-loop feedback is to reduce SINS error divergence, and the SINS solution values take into account the estimated output values of the filter as well as errors such as zero bias of the IMU [12]. Therefore, the solution value of the SINS system is the final output value of the combined system, which is also the optimal estimation result of the filter. Then, at the end of the current calculation cycle, the state estimator of the parameter error of the previous cycle in the filter is finally reset and the next filtering estimation will be performed. The integrated navigation system designed in this paper is a SINS/GPS tightly coupled system based on MEMS inertial sensors, and the pseudorange and pseudorange rate information derived using this method have significant variances when the carrier is in a highly dynamic environment, which has an impact on the output of the integrated filter. In addition, the system model is more difficult to determine when the initial unfolding point has large attitude and position errors. Therefore, the strongly linked closed-loop feedback mode is used for the integrated navigation system in this research. Firstly, in order to complete the acquisition of the initial navigation parameters for the SINS platform, the GPS receiver measures the pseudorange and pseudorange rate while the gyroscope and accelerometer obtain the angular velocity and acceleration information of the moving carrier [13]. Next, the input of the combined system filter is constructed by the pseudorange and pseudorange rate data computed by SINS, with the pseudorange and pseudorange rate

data provided by GPS. Finally, the estimation error was fed back to the SINS solver loop in real time to obtain real-time high-precision attitude, velocity, and position information.

## 2. Materials and Methods

### 2.1. Implementation Principle of VB–CKF Algorithm

Given that the tightly coupled model proposed during this research has nonlinear properties and that the CKF is a suboptimal filter based on the theoretical foundation of Bayesian filtering, its system model frequently has the following structure:

$$\begin{cases} x_k = f(x_{k-1}) + w_{k-1} \\ y_k = h(x_k) + v_k \end{cases} \tag{1}$$

In Equation (1), $x_k$ and $y_k$ are the n-dimensional state vector and L-dimensional measurement vector of the system, $w_k$ and $v_k$ are the system noise and measurement noise, respectively, $f$ and $h$ are the nonlinear state model and nonlinear measurement model of the system [14].

Then, the derivation defines that $Y = \{y_1, y_2, \cdots y_k\}$ is the set of measurements from the initial moment to moment K. According to reference [15], the core idea of Bayesian filtering is that, after the state posterior probability density function $P(x_k|Y_k)$ at moment k is known, the state posterior probability density function at the next moment $P(x_{k+1}|Y_{k+1})$ can be updated by using the measurement value $z_{k+1}$ at the next moment in the process of time updating and measurement updating. In the time-updating step, the one-step state prediction probability density function $p(x_{k+1}|Y_k)$ is computed by the integral of the following equation:

$$p(x_{k+1}|Y_k) = \int p(x_k|Y_k)p(x_{k+1}|x_k)dx_k \tag{2}$$

Finally, the expression $p(x_{k+1}|Y_k)$ can be obtained as:

$$\begin{cases} p(x_{k+1}|Y_{k+1}) = \frac{1}{C_{k+1}}p(x_{k+1}|Y_k)p(y_{k+1}|x_{k+1}) \\ C_{k+1} = \int p(x_{k+1}|Y_k)p(y_{k+1}|x_{k+1})dx_{k+1} \end{cases} \tag{3}$$

In Equations (2) and (3), above, $p(x_{k+1}|x_k)$ is one-step state transition probability density and $p(y_{k+1}|x_{k+1})$ is a measure of the probability density function.

Considering that both the measurement equation and the state equation are nonlinear modes at this moment and the noise is Gaussian white noise, the state prediction matrix and the prediction error covariance matrix should be modified to the nonlinear state first:

$$\begin{cases} \hat{x}_{k|k-1} = \int f_{k-1}(x_{k-1})F(x_{k-1}, \hat{x}_{k-1}, P_{k-1})dx_{k-1} \\ P_{k|k-1} = \int \left(f_{k-1}(x_{k-1}) - \hat{x}_{k|k-1}\right)\left(f_{k-1}(x_{k-1}) - \hat{x}_{k|k-1}\right)^T F(x_{k-1}, \hat{x}_{k-1}, P_{k-1})dx_{k-1} \\ \quad = \int f_{k-1}(x_{k-1})f_{k-1}(x_{k-1})^T F(x_{k-1}, \hat{x}_{k-1}, P_{k-1})dx_{k-1} - \hat{x}_{k|k-1}\hat{x}_{k|k-1}^T + Q_{k-1} \end{cases} \tag{4}$$

The measurement equation's $(y_k = h_k(x_k) + v_k)$ mean, variance, and cross-covariance matrix take on the following form:

$$\begin{cases} y_{k|k-1} = \int h_k(x_k)F(x_k, \hat{x}_{k-1}, P_{k-1})dx_k \\ P_{k|k-1} = \int \left(h_k(x_k) - \hat{y}_{k|k-1}\right)\left(h_k(x_k) - \hat{y}_{k|k-1}\right)^T F(x_k, \hat{x}_{k-1}, P_{k-1})dx_k \\ \quad = \int h_k(x_k)h_k(x_k)^T F(x_k, \hat{x}_{k-1}, P_{k-1})dx_k - \hat{y}_{k|k-1}\hat{y}_{k|k-1}^T + \hat{R}_k \\ P_{x_ky_k} = \int \left(x_k - \hat{x}_{k|k-1}\right)\left(h_k(x_k) - \hat{y}_{k|k-1}\right)^T F(x_k, \hat{x}_{k-1}, P_{k-1})dx_k \\ \quad = \int x_k h_k(x_k)^T F(x_k, \hat{x}_{k-1}, P_{k-1})dx_k - \hat{x}_{k|k-1}\hat{y}_{k|k-1}^T \end{cases} \tag{5}$$

The forms of state estimation and covariance estimation are now expressed as:

$$\begin{cases} K_k = P_{x_k y_k} P_{y_{k|k-1}}^{-1} \\ P_k = P_{k|k-1} - K_k P_{y_k} K_K^{\mathrm{T}} \\ \hat{x}_k = \hat{x}_{k|k-1} + K_k \left( y_k - \hat{y}_{k|k-1} \right) \end{cases} \tag{6}$$

In the above equation, $P_k$ represents the measurement noise variance, $\hat{x}_{k|k-1}$ represents the first- and second-order matrix expression of the predicted distribution function, and $\hat{x}$ represents the mean value.

According to Equations (4) and (5), it is found that to implement nonlinear Gaussian filtering, we can calculate multiple integrals of the nonlinear function $\times$ Gaussian probability density function, which is [16]:

$$I(f) = \int f(x) N(x, \hat{x}, P_x) dx \tag{7}$$

Using the sphere-radial cubature criterion, the above equation can be approximated as follows:

$$I(f) \approx \frac{1}{2n} \sum_{i=1}^{2n} f(\sqrt{P_x} \zeta_i + \hat{x}) \tag{8}$$

where $\zeta_i$ is $n$ column vectors, where these $n$ column vectors form the cubature points matrix, and $2n$ of these points is denoted as [17]:

$$\chi_i = \sqrt{P_x} \zeta_i + \hat{x} \tag{9}$$

Finally, the mean, variance, and cross-covariance of the measurement equation can be approximately expressed in analytical form as follows:

$$\begin{cases} \hat{y} = \frac{1}{2n} \sum\limits_{i=1}^{2n} f(\sqrt{P_x} \zeta_i + \hat{x}) \\ P_y = \frac{1}{2n} \sum\limits_{i=1}^{2n} f(\sqrt{P_x} \zeta_i + \hat{x}) f(\sqrt{P_x} \zeta_i + \hat{x})^T - \hat{y}\hat{y}^T \\ P_{xy} = \frac{1}{2n} \sum\limits_{i=1}^{2n} (\sqrt{P_x} \zeta_i + \hat{x}) f(\sqrt{P_x} \zeta_i + \hat{x})^T - \hat{x}\hat{y}^T \end{cases} \tag{10}$$

Since $2n$ points are taken in this experiment, the weight $\omega = 1/2n$. This weight is always positive, so the numerical divergence problem is avoided.

In the system state equation $\dot{x} = f(x) + w$ of this SINS/GPS tightly coupled model, x = [$\delta q_0 \delta q_1 \delta q_2 \delta q_3 \delta V_E \delta V_N \delta V_U \, \delta L \delta \lambda \delta h \varepsilon x \, \varepsilon y \, \varepsilon z \, \nabla x \nabla y \nabla z b_t d_t$], attitude errors $\delta q_0, \delta q_1, \delta q_2, \delta q_3$ are the main factors leading to the system accuracy offset. $\delta V_E$, $\delta V_N$, and $\delta V_U$ are the amount of velocity error in three directions, $\delta L$, $\delta \lambda$, and $\delta h$ are the amount of position error, $\varepsilon x$, $\varepsilon y$, and $\varepsilon z$ represent random constant value drift of the gyro, $\nabla x$, $\nabla y$, and $\nabla z$ represent random constant drift of the accelerometer, and $b_t$ and $d_t$ denote the clock offset and drift of the receiver.

In this experiment, the difference between the pseudoranges and pseudorange rate provided by GPS and those calculated by SINS are used as the observed quantities [18]:

$$\begin{cases} \delta p^k = \sqrt{(x_l - x_k)^2 + (y_l - y_k)^2 + (z_l - z_k)^2} - \sqrt{(x - x_k)^2 + (y - y_k)^2 + (z - z_k)^2} - dt - \bar{\varepsilon}_p^k \\ \delta \dot{p}^k = u_{lx}^k(x_l - x_k) + u_{ly}^k(y_l - y_k) + u_{lz}^k(z_l - z_k) - u_x^k(x_l - x_k) - u_y^k(y_l - y_k) - u_z^k(z_l - z_k) - dt - \dot{\varepsilon}_p^k \end{cases} \tag{11}$$

In the above equation, $\delta p^k$ and $\delta \dot{p}^k$ represent the pseudorange measurement error and pseudorange rate measurement error of the KTH satellite, respectively. $[x_k y_k z_k]$ is the location of the satellite under an Earth-Centered Earth-Fixed (ECEF) system, $[xyz]$ is the actual distance from the satellite to the carrier, and the position of the SINS carrier is

$[x_l y_l z_l]$. $[u_{lx}^k u_{ly}^k u_{lz}^k]$ is the unit observation vector on the SINS of the KTH satellite under ECEF system.

Variational Bayesian (VB) is a method for calculating high-order integrals, which is usually used in complex models of observed and unobserved variables [19]. In order to achieve the estimated distribution of non-linear functions with fewer errors, it makes the assumption that the likelihood function, likewise, follows a Gaussian distribution and sets a dynamic threshold for the marginal likelihood function. The marginal likelihood value of the model can be used to determine how well the model fits the data; the higher the value, the better the fit, and the lower the error will be. For the simultaneous estimate of state and measurement noise variance, Wang ShiYuan introduced the VB–KF algorithm [20]; however, its underlying filter is the KF algorithm, which is not appropriate for nonlinear systems. As a result, we suggest the VB–CKF combination technique in this study to enhance the adaptability of changes in noise statistics in a tightly linked SINS/GPS system.

In general, the normal distribution $N(\mu, \sigma^2)$ represents the distribution relationship of multiple measurements of a certain system state quantity. The mean $\mu$ of the population can be used to evaluate the sample variance model for the state model that follows the normal distribution. If $\mu$ is determined, the sample variance model follows an inverse Gamma distribution with conjugation. Therefore, when VB is used for error estimation, the prior probability density function and posterior probability density function have similar expressions that can be written as $IG(\sigma^2, \alpha, \beta)$, where $\alpha$ and $\beta$ represent the parameters to be calculated. $E(\sigma^2) = \beta/\alpha$ can be expressed from the means of inverse Gamma, and it can be known that the noise variance $\sigma^2$ can be determined when $\alpha$ and $\beta$ are obtained. VB is an approximation method, which can solve a more complex posterior distribution by approximating several known distributions. Therefore, the posterior estimation of $IG\left(\sigma_{k,i}^2, \alpha_{k,i}, \beta_{k,i}\right)$ can be obtained by using the VB estimator. Since the variance parameters $\alpha_k$ and $\beta_k$ are only related to the variance parameters $\alpha_{k-1}$ and $\beta_{k-1}$ at the previous time, a variance change coefficient $\rho$ at the adjacent time is set to represent the relationship between them. Therefore, the prediction equation can be calculated using the statistical linear method.

For the posterior update of the noise parameters, it is first necessary to obtain the predicted residual vector at the current moment, and then the approximate measurement noise can be calculated based on the residual $C_{y_k}$. Finally, the noise parameter $\beta_k$ can be updated, which is [21]:

$$\beta_k = \beta_k^- + \frac{1}{2}(y_k - \hat{y}_k)^2 + \frac{1}{2} diag\left(C_{y_k}\right) \tag{12}$$

For Equation (12), the state vectors $\hat{y}_k$ and $C_{y_k}$ can be approximated by collecting sample points for the weighted summation operation and then using the filter estimates $x_k$ and $P_k$. It is known that the CKF filter is an approximation of the statistical linearization filtering, and the conclusion of reference [22] shows that statistical linearization by using the estimated value of filtering depends on the estimation accuracy of nonlinear filtering. So, the estimation results of filtering can be used for statistical linearization, and its expression is:

$$\begin{cases} y_k = A_{h,k} x_k + B_{h,k} + d_{h,k} + v_k \\ A_{h,k} = P_{x_k y_k}^T P_{k|k-1}^{-1} \\ B_{h,k} = \hat{y}_{k|k-1} - A_{h,k} \hat{x}_{k|k-1} \end{cases} \tag{13}$$

In the equation, $A_{h,k}$ and $B_{h,k}$ represent the statistically linearized state transfer matrix and the matrix composed of vectors unrelated to the state transfer matrix, respectively. $d_{h,k}$ represents the statistical linearization error, and its variance can be represented by $C_{y_{k|k-1}} - A_{h,k} P_{k|k-1} A_{h,k}^T$. $C_{y_{k|k-1}}$ represents the prediction error covariance matrix of the

homogeneous measurement equation, combining the above-generalized equation; the formula for $C_{y_{k|k-1}}$ can be obtained as follows:

$$C_{y_{k|k-1}} = \int \left(h_k(x_k) - \hat{y}_{k|k-1}\right)\left(h_k(x_k) - \hat{y}_{k|k-1}\right)^T F(x_k, \hat{x}_{k-1}, P_{k-1})dx_k \tag{14}$$

The predicted mean and covariance in the measurement equation can be expressed after statistical linearization [23] as:

$$\begin{cases} y'_{k|k-1} = A_{h,k}\hat{x}_{k-1} + B_{h,k} \\ P'_{k|k-1} = A_{h,k}P_{k|k-1}\left(C_{y_{k|k-1}} - A_{h,k}^T\right) + R_k \end{cases} \tag{15}$$

According to reference [24], the variance parameter can only be estimated after updating the filtering posterior, so the incomplete sequence after statistical linearization can be obtained from the filtered results $\hat{x}_k$ and $P_k$ as follows:

$$y_k - \hat{y}'_k = y_k - (A_{h,k}\hat{x}_k + B_{h,k}) \tag{16}$$

Similarly, the approximate measurement noise correction term $C'_{y_k}$ after statistical linearization can be expressed as:

$$C'_{y_k} = A_{h,k}P_k A_{h,k}^T + C_{y_{k|k-1}} - A_{h,k}P_{k|k-1}A_{h,k}^T \tag{17}$$

From Equations (12), (16), and (17), the posterior update formula for $\beta_k$ can be determined as:

$$\beta_k = \beta_k^- + \frac{1}{2}(y_k - A_{h,k}\hat{x}_k + B_{h,k})^2 + \frac{1}{2}diag\left(A_{h,k}P_k A_{h,k}^T + C_{y_{k|k-1}} - A_{h,k}P_{k|k-1}A_{h,k}^T\right) \tag{18}$$

The above equation is the core of the VB noise estimator with statistical linearization. The two terms $B_{h,k}$ and variance $C_{y_{k|k-1}} - A_{h,k}P_{k|k-1}A_{h,k}^T$ ensure the stability when estimating the noise variance and improve the estimation accuracy.

### 2.2. Algorithm Test Plan

By combining the VB estimator mentioned above with the filter based on the CKF algorithm, the specific implementation process of the VB–CKF algorithm can be concluded as follows:

1.  Time update

    (1)  Calculate the cubature points near the state variable that generate propagation with the state equation [25]:

    $$\begin{cases} \chi_{k-1} = \hat{x}_{k-1} + \left(\sqrt{nP_{k-1}}\right)_i \hat{x}_{k-1} - \left(\sqrt{nP_{k-1}}\right)_i \\ \chi_{k|k-1} = f_{k-1}(\chi_{k-1}) \end{cases} \tag{19}$$

    (2)  The covariance matrix and the predicted value of the measurement noise variance parameter are as follows:

    $$\hat{x}_{k|k-1} = w\sum_{i=1}^{2n} \chi_{i,k|k-1} \tag{20}$$

    $$P_{k|k-1} = w\sum_{i=1}^{2n} \chi_{i,k|k-1}\chi_{i,k|k-1}^T - \hat{x}_{k|k-1}\hat{x}_{k|k-1}^T + Q_{k-1} \tag{21}$$

    $$\alpha_{k|k-1} = \rho\alpha_{k-1} \qquad \beta_{k|k-1} = \rho\beta_{k-1} \tag{22}$$

2.  Parameter passing:

    $$\alpha_k = \alpha_{k|k-1} \qquad \beta_k = \beta_{k|k-1} \tag{23}$$

3. Measurement update:

(1) Analyze the data and calculate the estimated variance of measurement noise:

$$\hat{R}_k = diag(\beta_k / \alpha_k) \tag{24}$$

(2) Generate cubature points propagating with the measurement equation near the state value $\hat{x}_{k|k-1}$:

$$\widetilde{\chi}_{k-1} = \hat{x}_{k|k-1} + \left(\sqrt{nP_{k|k-1}}\right)_i \hat{x}_{k-1} - \left(\sqrt{nP_{k|k-1}}\right)_i \tag{25}$$

$$\gamma_{k|k-1} = h_k(\hat{\chi}_{k-1}) \tag{26}$$

(3) Calculate the predicted values of measurement update, autocovariance matrix, and cross-covariance matrix, respectively [26];

$$P_{y_k} = w \sum_{i=1}^{2n} \gamma_{i,k|k-1} \gamma_{i,k|k-1}^T - \hat{y}_{k|k-1} \hat{y}_{k|k-1}^T + R_k \tag{27}$$

$$\gamma_{k|k-1} = h_k(\hat{\chi}_{k-1}) \tag{28}$$

$$P_{x_k y_k} = w \sum_{i=1}^{2n} \hat{\chi}_{i,k|k-1} \gamma_{i,k|k-1}^T - \hat{x}_{k|k-1} \hat{y}_{k|k-1}^T \tag{29}$$

(4) Calculate the filtering gain, state estimation value, and covariance matrix, respectively [27];

$$K_k = P_{x_k y_k} P_{y_k}^{-1} \tag{30}$$

$$P_{k|k-1} = w \sum_{i=1}^{2n} \chi_{i,k|k-1} \chi_{i,k|k-1}^T - \hat{x}_{k|k-1} \hat{x}_{k|k-1}^T + Q_{k-1} \tag{31}$$

$$\hat{x}_k = \hat{x}_{k|k-1} + K_k \left( y_k - \hat{y}_{k|k-1} \right) \tag{32}$$

4. Use the filtering results $\hat{x}_k$ and $P_k$ to obtain the values of $A_{h,k}$, $B_{h,k}$, and $C_{y_{k|k-1}} - A_{h,k} P_{k|k-1} A_{h,k}^T$ after statistical linearization;

5. Finally, $A_{h,k}$, $B_{h,k}$, and $C_{y_{k|k-1}} - A_{h,k} P_{k|k-1} A_{h,k}^T$ are substituted into Equation (18) to obtain the updated value of $\beta_k$ [28].

Both prediction and updating procedures are included in the measurement noise variance in VB–CKF. This algorithm uses estimation of the previous moment to pass instead, and the current estimate is only related to the previous estimate. This method avoids the storage and summation operation of the new interest sequence in the sliding window in the conventional method because all of the variance of the measurement noise is unknown. Additionally, the noise-tracking ability, which fluctuates independently in the covariance matrix, can be adjusted separately by modifying the weight $\rho$. From (15), it can be seen that the parameter to be estimated is only related to $\rho$, which is usually $\rho \in (0,1]$. A lower value of $\rho$ means that the predicted value of the variance parameter is less related to the previous time, and the actual measurement noise changes more sharply. Therefore, the $\rho$ value can be adjusted according to different application environments to flexibly enhance the adaptive capability of the system.

Figure 2 depicts the VB–CKF algorithm's adaptive tight coupling fusion process in combinatorial navigation systems. The technique first estimates the closely coupled state with CKF and then simultaneously estimates the variance of measurement noise from satellites using VB. The SINS solution value is corrected using the estimated state feedback, the CKF filtering is implemented using the calculated variance parameter, and, finally, the forecast value is updated. The predicted value is estimated and updated, as well as the

subsequent sample point during this processing [29]. The system also outputs the filtered navigation parameter data at the same time.

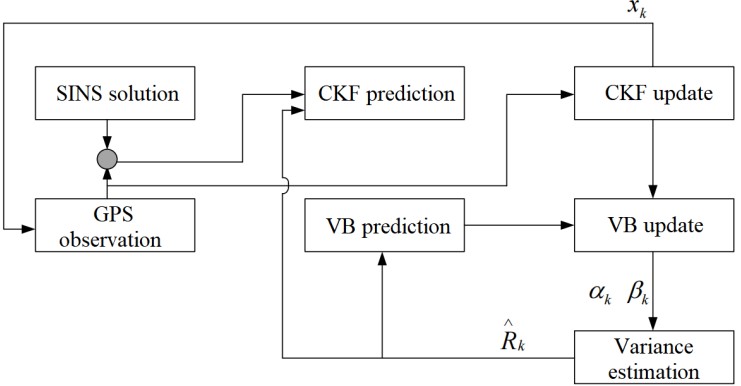

**Figure 2.** VB–CKF algorithm adaptive fusion flow chart.

First, the carrier's trajectory was simulated using the trajectory generator on the MATLAB R2019b software platform. The platform also featured a VB–CKF tightly coupled algorithm, a GPS data generator, and an IMU data generator. Software tools that can simulate actual devices in the MATLAB platform include the trajectory generator, IMU data generator, and GPS data generator. For the purpose of solving SINS, the IMU data generator can simulate the actual gyro and acceleration measurements. The IMU is configured as MEMS grade with low precision in this experiment, and the solving frequency is 10 Hz.

Following the selection of the IMU type, which is MEMS-level and has a low degree of accuracy, the following error settings are specified for the IMU's gyroscope and accelerometer: the constant drift of the gyroscope is set at $10°/h$, the gyroscope white noise is $0.05°/\sqrt{h}$, and the gyroscope cross-coupling error is 200 ppm. The accelerometer bias is set to 1 mg, the accelerometer white noise is $0.5 \text{ mg}/\sqrt{Hz}$, and the accelerometer cross-coupling error is 200 ppm.

GPS data generator can be used to acquire the test object's pseudorange and pseudorange rate. Other settings are set as follows after selecting the week_824 file as the satellite book during the simulation and setting the output frequency to 1 Hz: the clock drift white noise is set to 0.1 m/s, the clock offset white noise is 10 m, the initial pseudorange value is set to 10 m, and the initial pseudorange rate measurement noise variance is 0.3 m/s [30]. As electromagnetic transmissions, GPS signal must account for atmospheric delay errors. Considering that the electromagnetic wave can produce bending and delay in the troposphere, it is necessary to model the tropospheric zenith delay ($TZD$) [31]. The modeling form given by the mapping function in this study is as follows:

$$TZD = \frac{0.002277}{f(\phi,h)} \times \left[ P_s + (\frac{1255}{T_s} + 0.05)e_s \right] \tag{33}$$

$$e_s = rh \times 6.11 \times 10^{\frac{7.5(T_s - 273.15)}{T_s}} \tag{34}$$

$$f(\phi,h) = 1 - 0.00266\cos(2\varphi) - 0.00028h \tag{35}$$

In the above three equations, $T_S$ is the surface temperature (K), $P_S$ is the surface air pressure (mbar), $e_S$ is the surface water pressure, and $rh$ is the surface relative humidity. $f(\phi,h)$ is a function of latitude and height, reflecting the variation in gravity acceleration with geographical location and altitude, $\phi$ is the geocentric geodetic latitude of the station, and $h$ is the geodetic height of the station.

In addition, when the electromagnetic wave signal passes through the ionosphere at a height of 60–1000 km, its propagation speed is changed with the change in electron concentration. According to the relationship between group refractive index and phase refractive

index in electromagnetic wave theory, the ionospheric pseudorange delay correction model from satellite s to observer o can be written as follows:

$$\Delta\rho = \int_s^o (n_{gr} - 1)ds = \int_s^o \frac{1}{2}\frac{f_p^2}{f^2}ds = \int_s^o \frac{1}{8\pi^2}\frac{d_e e_0^2}{f^2 m_e \varepsilon_0}ds = \frac{40.3}{f^2}TEC \tag{36}$$

where *TEC* is the propagation path of the signal from satellite *s* to observation point *o*, and its expression can be set as follows:

$$TEC = \int_s^o d_e(s)ds \tag{37}$$

The experiment selects longitude $\lambda = 106.61$, latitude $L = 29.53°$, and altitude $H = 450$ m as the initial position. After setting the initial parameters in the system, in order to verify the adaptability of the algorithm, it is necessary to adjust the measurement noise in the simulation movement of the carrier, especially to make the GPS signal change rapidly with time in a certain period of time, but the specific situation of the change is random. Therefore, combining Equation (14), the tracking ability of the system is enhanced by adjusting the initial weight $\rho = 1 - e^{-2}$. In order to ensure that the error of VB–CKF does not increase with time under this condition [32], the number of cubature points is set to 1600, and the initial value of the filter can be set as $x_0 = [0_{18\times1}]$.

## 3. Results

After the above simulation process, the simulation results are shown in Figure 3.

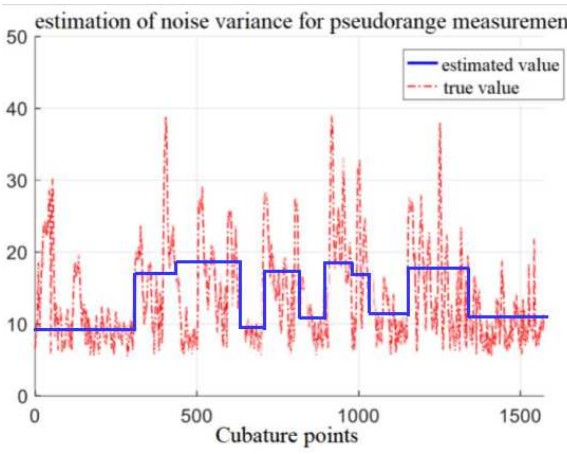

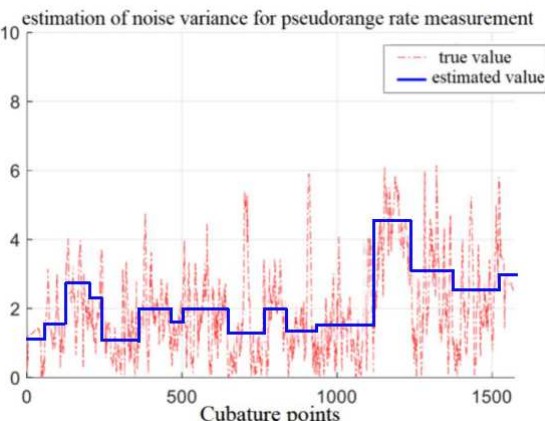

**Figure 3.** VB–CKF pseudorange and pseudorange rate measure noise variance estimation under the condition of drastic noise variance.

Equation (24), whose value is influenced by the noise variance of neighboring moments, can be used to estimate the value of the corresponding measurement noise when the equivalent measurement noise swings significantly in a short amount of time, as shown in Figure 3. The estimated value will change if the noise variance does. Even though there will be modifications, there is a general inhibitory impact. The VB–CKF algorithm has good flexibility, as can be observed from the overall simulation results; when the noise variance diverges, the system can quickly suppress it. We can still see from the image that the algorithm has a decent chasing ability, and the data results are ideal, despite the fact that the noise variance changes rapidly in a short amount of time and some cubature points are not recorded in the matrix for the parameter transfer. Therefore, as a whole, the VB–CKF algorithm has good tracking ability in both cases where the noise variance is constant and rapidly changing, and there is no residual record in the entire system. Figure 3 compares the simulation result curves for the velocity, position, and attitude parameter estimation error of the CKF algorithm with the VB–CKF algorithm under the aforementioned simulated settings.

As shown in Figure 4, the CKF algorithm and VB–CKF algorithm have estimation error convergence within the ideal range when the variation range of system measurement noise is within the controllable range, but the VB–CKF algorithm has a relatively faster convergence speed and performs better in terms of speed, position, and tracking accuracy. In the attitude calculation, because the commercial IMU is selected, the performance is not as good as the military level in all aspects. When the parameters are set, the compensation accuracy is not enough, which leads to a great impact on the gyro accuracy. When the CKF filter is used for simulation, the attitude angle curve produces slightly significant burrs.

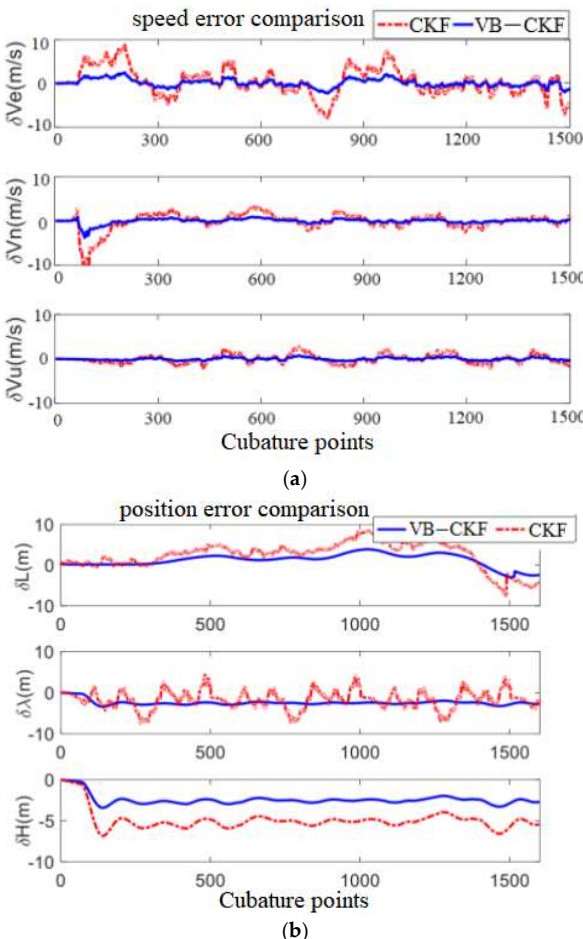

**Figure 4.** *Cont.*

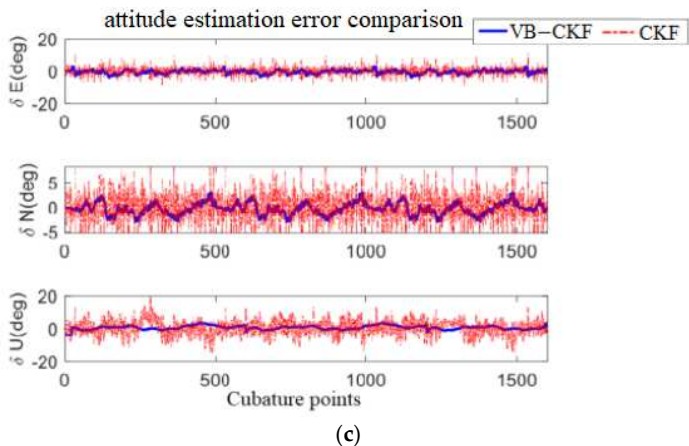

(c)

**Figure 4.** Comparison of speed, position, and estimation error of the two algorithms under the condition of drastic noise variance. (**a**) Speed estimation error comparison. (**b**) Comparison of position estimation errors. (**c**) Comparison of attitude estimation errors.

On the whole, the CKF algorithm and VB–CKF algorithm have ideal estimation accuracy of attitude angle, and there is no obvious divergence. However, from the point of view of details, VB–CKF has a better effect, and this algorithm can also produce an obvious response to noise mutation occurring in a short time. Compared to the CKF algorithm, the VB–CKF algorithm weighs various parameters, which makes it more adaptive. The observation results show that the diagonal elements of the time-varying noise variance matrix have the function of independent adjustment and tracking, and the estimation accuracy has been greatly improved. The estimation error of the VB–CKF algorithm is shown in Table 1. Compared with the CKF algorithm, the VB–CKF algorithm improves attitude angle accuracy by around 30% while reducing speed and position estimate error (RMSE) by about 45%.

**Table 1.** Comparison of estimation errors (RMSE) among tightly coupled algorithms.

| Error | Direction | CKF | VB–CKF |
|---|---|---|---|
| | East (E) | 1.679 | 0.914 |
| speed error (m/s) | North (N) | 3.124 | 2.281 |
| | Up (U) | 2.772 | 1.751 |
| | Latitude (L) | 2.572 | 1.173 |
| position error (m) | Longitude ($\lambda$) | 2.143 | 0.995 |
| | Height (H) | 2.674 | 1.680 |
| | East (E) | 1.646 | 1.169 |
| attitude error (deg) | North (N) | 1.748 | 1.183 |
| | Up (U) | 1.467 | 1.130 |

In order to verify the fusion performance of the VB–CKF algorithm in the combined system, the experimental sites are chosen in the outdoor playground with strong GPS signal and complicated portions with significant GPS signal occlusion in some road sections, as shown in Figure 5. The system prototype consists of SINS developed independently by our experimental team and the NEO-7N-0-00 GPS module produced by ublox company. In order to increase the system's stability, the test team should secure the prototype to the waist and conduct tests in the selected test site to verify the performance of the combined system, as shown in Figure 6.

The outdoor playground is located in an open area without occlusion around it, and the system can receive a stable GPS signal (in the whole test process, the number of visible satellites that can be stably received by the GPS module reaches six). Therefore, the initial value of the weight of the parameter transfer equation in the VB–CKF algorithm is set to $\rho = 1 - e^{-5}$. The movement path of the tester is shown in Figure 5a as A-B-C-D-E-A

(starting from point A, two different movement modes of walking and accelerating running can be switched arbitrarily according to the subjective will of the tester and finally stop at point C and save the data in the whole process). Three groups of experiments were carried out using the pure SNIS algorithm, CKF algorithm, and VB–CKF algorithm, respectively. The following is the estimated reproduction map of different navigation algorithms and the final heading angle error comparison table (Figure 7 and Table 2).

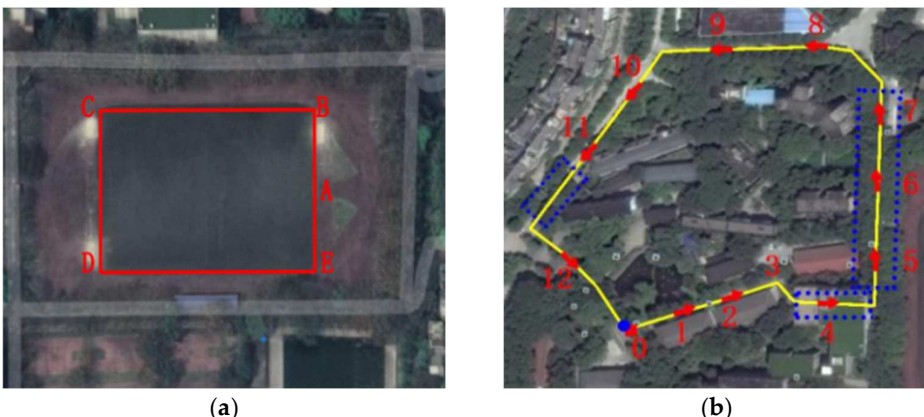

|  |  |
|:-:|:-:|
| (**a**) | (**b**) |

**Figure 5.** Two types of experimental sites: (**a**) outdoor playground with experimental path of A-B-C-D-E-A, (**b**) complex sections with marked numbers 0–12, which are the experimental routes.

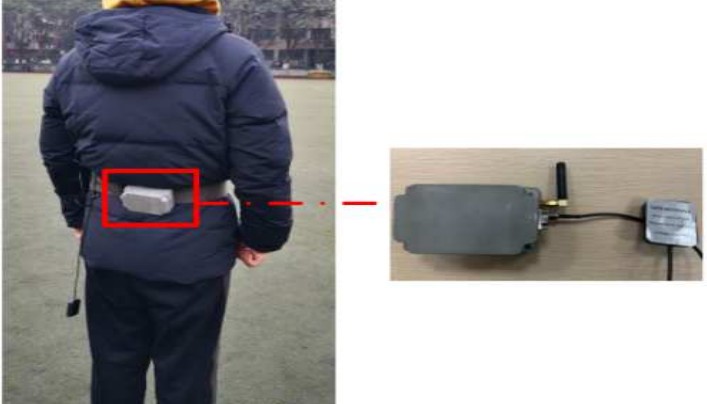

**Figure 6.** Integrated navigation system prototype display diagram and wearing diagram.

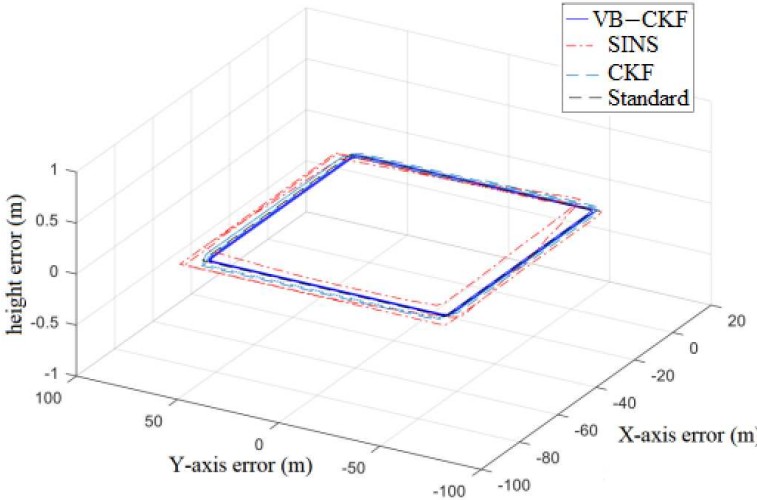

**Figure 7.** Comparison of trajectory reproduction of different algorithms in playground road sections.

**Table 2.** Table of test results for different testers.

| Number | Initial Heading/° | Final Heading (SINS/CKF/VB–CKF) | Error (SINS/CKF/VB–CKF) |
|---|---|---|---|
| 1 | 0.51 | 15.54/2.07/1.8 | 15.03/1.51/1.24 |
| 2 | 2.65 | 16.35/3.97/3.7 | 14.7/1.32/1.05 |
| 3 | 354.65 | 11.32/356.23/355.8 | 16.67/1.58/1.15 |
| 4 | 358.27 | 15.39/0.53/0.17 | 17.11/2.26/1.90 |
| 5 | 357.21 | 14.86/359.53/359.03 | 17.65/2.32/1.82 |

The results show that due to the open and flat terrain, the visible number of satellites is high, and the GPS data almost coincide with the standard line. The VB–CKF algorithm filter focuses more on the GPS measurements, and the VB estimator compensates for the SINS noise characteristics, so the combined VB–CKF-based system has optimal positioning results compared to the other two algorithms. When the GPS measurement noise is stable (the number of GPS observables is greater than 4), the sum of the absolute values of the errors of the VB–CKF algorithm in the two-dimensional plane is about 1 m, which is about 10% better than the CKF algorithm in terms of positioning accuracy. In the heading angle comparison, setting the direction pointed by AB as the initial heading 0°, the accuracy of the heading angle of the VB–CKF algorithm is improved by about 20% on average compared with that of the CKF algorithm.

The whole length of the mountain section is about 1050 m, and the GPS signal of some sections is seriously occluded, and the number of visible points is less than three, which is represented by the blue dashed line box in Figure 5b, accounting for about 30% of the total path. Interval 3–4 is the indoor interval. So, the initial value of the weight of the parameter transfer equation in the VB–CKF algorithm is set to $\rho = 1 - e^{-2}$ to reduce the influence of the measurement value in the system. The starting point of the tester is the position of point 0 in the figure, and the tester proceeds along the direction indicated by the arrow until he returns to the starting position of point 0 from point 12. The remaining test conditions are the same as the previous test. Figure 8 shows a reproduction of the motion trajectories of different algorithms in this environment.

It can be seen from Figure 8 that the CKF algorithm cannot adjust the noise statistical characteristics in real time, and there is a noise jump in the GPS measurement in this test, so that the system gain direction cannot be adjusted in time, resulting in a large closed-loop error of 4.6 m when finally returning to the origin, but the closed-loop error of the VB–CKF algorithm phase is only about 2.5 m. The height difference from the highest point to the lowest point of the CKF algorithm is about 18.9 m, and the final convergence error is about 2.5 m. The difference between the highest point and the lowest point collected by the prototype of the VB–CKF algorithm is about 17.2 m, and the height error after returning to the origin is about 1.5 m. Since the GPS measurement noise in this test environment varies drastically, the CKF is unable to adjust the estimator gain, which leads to a certain degree of deviation, while the prediction of the variance parameters $\alpha_k$, $\beta_k$ of the VB–CKF algorithm is only determined by the variance parameters $\alpha_{k-1}$, $\beta_{k-1}$ of the previous time and the variance variation coefficient $\rho$ of the adjacent time, which realizes the synchronous estimation of the combined system state and the GPS measurement noise variance, so the positioning accuracy of the VB–CKF algorithm is higher.

According to the results of multiple sets of experimental data in Figure 9, the positioning accuracy of the VB–CKF algorithm is about 38% higher than that of the CKF algorithm.

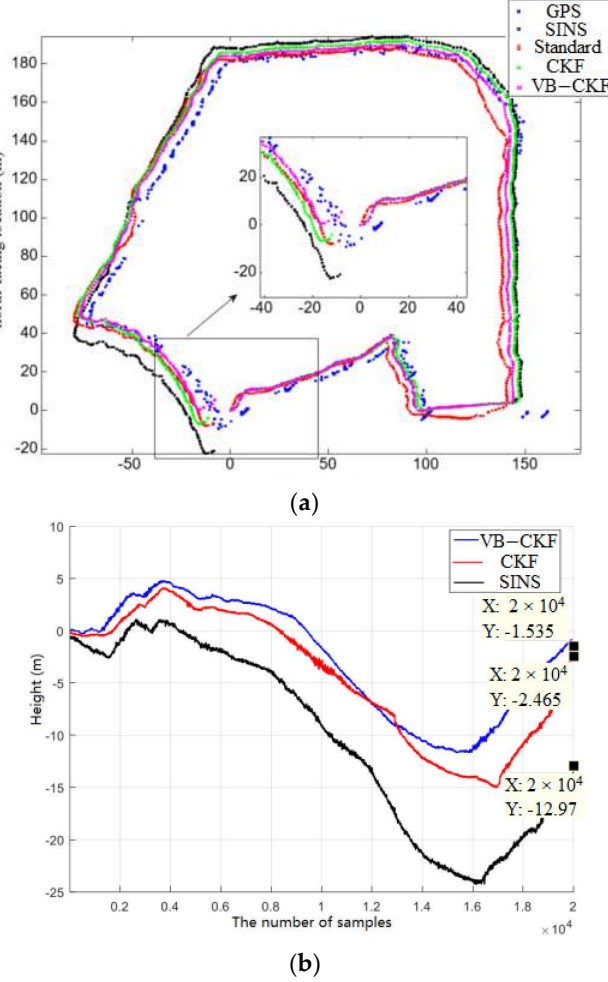

**Figure 8.** (**a**) Comparison of planar trajectory reproduction of different algorithms in complex road sections. (**b**) Comparison of height trajectory reproduction of different algorithms in complex road sections.

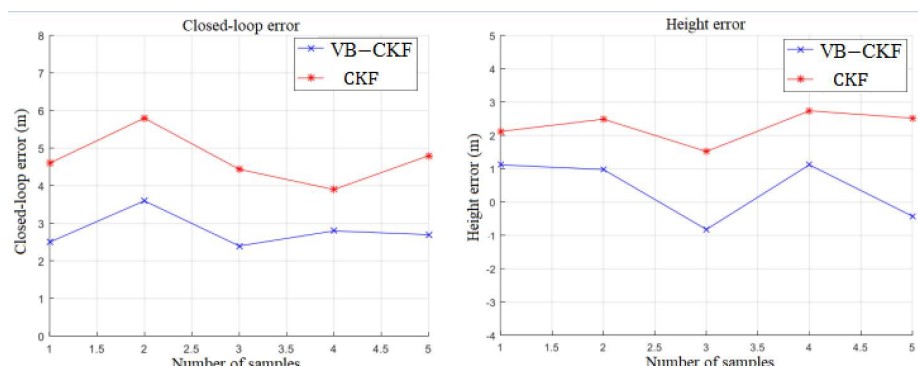

**Figure 9.** Comparison of multiple groups of test data.

## 4. Discussion

Modern navigation and positioning technology is widely employed in aerospace, land vehicles, and ocean cruise navigation and positioning as an essential cutting-edge technology in contemporary society [1]. Modern navigation and positioning technologies can be categorized as inertial positioning and navigation, satellite navigation, geomagnetic navigation, and environmental-assisted navigation [2], depending on how data from navigation sources are obtained. According to the benefits and drawbacks of various

positioning sources, integrated navigation systems—which combine two or more different positioning sources—have, in practice, taken over in terms of modern navigation and positioning technology, with SINS and GPS being the most popular combination [3].

Currently, there are two main categories of information-coupling methods: loose coupling and tight coupling. When compared to the loose coupling model, in which the systems operate independently, the tight coupling structure is a type of combination mode that allows SINS and GPS systems to modify one another. It is also more in line with the demands of contemporary society for high accuracy and high robustness in long-endurance positioning [4,13]. Many researchers have made in-depth studies on SINS and GPS information fusion to improve the long endurance accuracy and stability of integrated navigation. In order to obtain the nonlinear fusion results of signals in practical applications, many researchers have proposed the EKF algorithm based on the KF algorithm, which is simple to implement and has fast convergence speed [5,6]. J. L. Carssidis et al. proposed a new SINS/GPS combination method based on UKF [8], and H. U. Heo proposed a combination algorithm of IMM and UKF soon after [9]. The above studies improved the filtering effect of information fusion in the nonlinear state, but there are still shortcomings, such as large high-order error, large amount of calculation, and low real-time performance.

In the preliminary research, it was found that CKF follows the volume numerical integration principle, and the transition weights are only related to the state dimension. Its time complexity is lower than UKF, and it avoids the high-order divergence problem of EKF [7,10,11]. Additionally, it was discovered that using VB as a method for computing high-order integrals in complex measurement models or unmeasured models is possible through the process of learning from the Bayesian notion, which is the foundation of the KF algorithm [14,15]. Therefore, VB is used to realize the synchronous estimation of the state and the measurement noise variance [19]. Wang ShiYuan proposed a VB–KF algorithm for the simultaneous estimation of state and measurement noise variance [20], but this algorithm is not suitable for nonlinear environments. Finally, combined with the above problems in the previous research, the VB–CKF algorithm is proposed to solve the above shortcomings and improve the accuracy and stability of the long-endurance navigation system. After experimental verification, compared with the CKF algorithm, the VB–CKF algorithm proposed in this paper can improve the positioning accuracy by about 10% under a stable environment and about 38% under severe environmental changes. It improves the adaptability of the integrated navigation system and ensures the accuracy and stability of the long sailing time.

In addition to obtaining GPS information through pseudorange measurement, there is another way: carrier phase measurement. Carrier phase measurement estimates the distance by measuring the carrier phase difference between the receiver and the satellite, which is characterized by periodic changes and is relatively insensitive to multipath interference. In the following, we discuss the performance comparison between carrier phase measurement and pseudorange measurement, and we explain why we decided to use pseudorange measurement in this study.

The first step is to establish an equation for the carrier phase measurement: the principle of carrier phase ranging can be expressed by Equation (38), where r represents the geometric distance between the satellite and the receiver, $\lambda$ is the carrier wavelength, $\phi$ is the phase change of the satellite carrier signal from the satellite end to the receiver end, and $N$ is an unknown number, usually called integer ambiguity.

$$\phi = \lambda^{-1}r + N \tag{38}$$

Due to the influence of error factors, such as receiver clock difference, satellite clock difference, and atmospheric delay, the distance between satellite and receiver determined by carrier phase observation includes the above error terms in addition to the real geometric

distance. Combining the above error term into Equation (1), the following carrier phase observation equation is obtained.

$$\phi = \lambda^{-1}(r + c(\delta t_u - \delta t^s) - I + T) + N + \varepsilon_\phi \tag{39}$$

Here, $\delta t_u$ and $\delta t^s$ are the receiver and satellite clock errors, respectively, $I$ and $T$ are the ionospheric and tropospheric delays, respectively, and $\varepsilon_\phi$ is the carrier phase measurement noise.

The actual carrier phase $\phi$ extraction process is performed by adding the amount of carrier phase change measured via the carrier loop between each observation calendar to the initial value of the carrier phase. When the phase-locked loop (PLL) can track a satellite signal continuously and accurately, it is equivalent to recording the carrier phase change value generated by the distance change of the actual satellite signal during the observation interval time.

Finally, the carrier phase observation obtained at each observation time $i$ can be expressed as follows:

$$\widetilde{\phi}_i = N_0 + Int(\phi)_i + Fr(\phi)_i \tag{40}$$

In the above equation, $N_0$ represents the initial integer ambiguity, $Int(\phi)_i$ represents the integer part of the carrier phase variation during the observation interval, and $Fr(\phi)_i$ represents the fractional part of the carrier phase variation.

Next, the performance comparison of carrier phase measurement and pseudorange measurement for the VB–CKF algorithm proposed in this study will be experimentally verified. For the first time, the experimental site is chosen in a section with both an empty section and a covered area of tall buildings marked by the blue dashed box. This experiment is divided into three groups, and the pure SINS algorithm, the VB-CKF algorithm of pseudorange measurement, and the VB-CKF algorithm of carrier phase measurement are used as positioning algorithms in turn. The experimenter starts from point 1, completes the route along the number 1-2-3-4-5-6-5-2-1 in Figure 10, and returns to the origin. Figure 10 shows the experimental roadmap, and Figure 11 and Table 3 show the comparison map of the final positioning data.

**Table 3.** Table of test results for different algorithm types.

| Algorithm Type | Error Mean Value/m | Error Variance Value/m |
|---|---|---|
| SNIS | 2.7 | 6.4 |
| VB–CKF (pseudorange) | 0.8 | 0.42 |
| VB–CKF (carrier phase) | 0.6 | 0.37 |

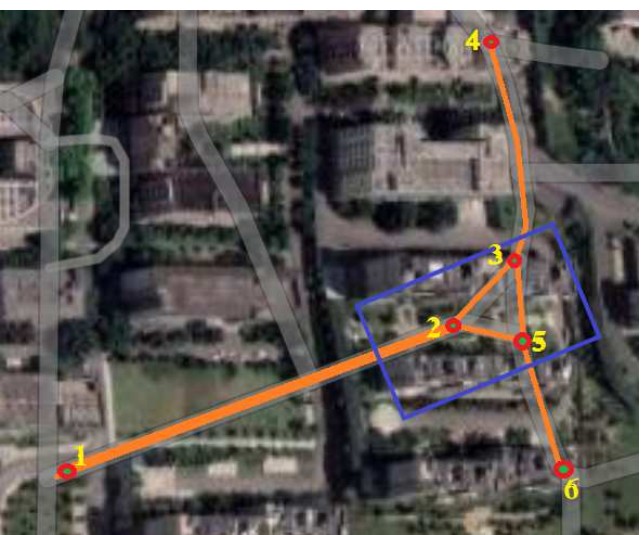

**Figure 10.** Experiment roadmap with marked numbers 1–6 as the experimental routes.

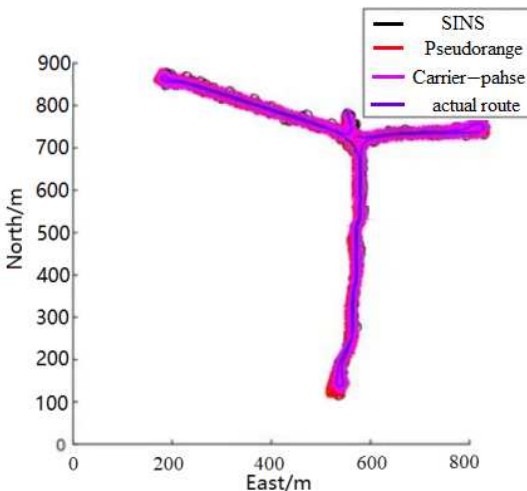

**Figure 11.** Comparison of positioning results.

The data in Figure 11 and Table 3 show that the VB–CKF algorithm based on pseudorange and the VB–CKF algorithm based on carrier positioning are close in terms of positioning effect. However, through the comparative analysis of the speed results in Figure 12 below, it can be seen that the number of points solved using the VB–CKF algorithm based on carrier positioning is significantly less than that solved using the VB–CKF algorithm based on pseudorange in the same time. This is because the carrier phase measurement has higher requirements and complexity for the accurate calculation of carrier phase and data differential processing, which requires more calculation and processing steps. Therefore, considering the real-time requirements of the solution data of the long-endurance integrated navigation system proposed in this study, and considering the hardware cost and practicability, this study chooses the pseudorandom measurement with higher cost performance.

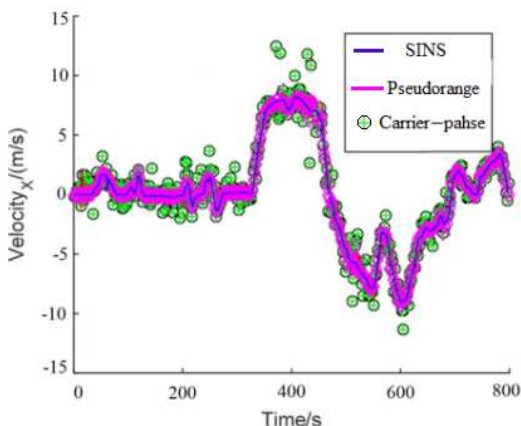

**Figure 12.** Comparison of the speed results.

Although the comparison of simulation data and numerous sets of experiments show that the VB–CKF method increased the navigation and positioning accuracy compared to other algorithms, there are still some areas that can be further improved in the follow-up research:

1. In this paper, motion patterns, such as jumping, bending forward, and crawling, are not taken into account, but these behavior patterns will appear multiple times in some actual complex scenes. The following study will take into account the aforementioned varied unorthodox motion modes in order to guarantee and enhance the integrated navigation system's long-endurance accuracy.

2. For the tightly coupled nonlinear data fusion problem, variational Bayesian filtering considers the propagation relationship between the time-varying characteristics of the noise parameters and the previous time, which provides a solution in practical application. However, at the simulation level, higher-order models such as ARMA in pattern recognition can be further used to deal with the time-series relationship of SINS/GPS measurement noise.
3. In order to accomplish long-term seamless positioning in intricate urban environments, further navigation information sources and positioning technologies should be added to the integrated navigation system, if this study is to continue to be useful in the application of smart cities. The latter study will concentrate on WIFI, navigation databases, and signal quality control criteria [33,34].

## 5. Conclusions

The SINS/GPS integrated system requires the fusion of navigation source information. During the fusion process, non-stationary noise will be generated over time, which reduces the stability and adaptability of the system.

(1) In this paper, a fusion method based on VB estimation is proposed, the CKF algorithm is used as a filter to deduce the VB–CKF algorithm in detail, and the whole process of the algorithm is introduced. The simulation results show that the VB–CKF algorithm performs better than the previous CKF algorithm in speed, position, and tracking accuracy.
(2) Through simulation and outdoor experiments in different scenarios, it is found that the VB–CKF algorithm is better than the CKF algorithm in terms of speed, position, and attitude errors. In the open area with good GPS signals, compared with the CKF algorithm, the VB–CKF algorithm improves the positioning accuracy by about 10% and improves the heading angle accuracy by about 20% on average; in complex long-distance road sections where GPS signal is partially inhibited, the VB–CKF algorithm improves the location accuracy by about 38% compared with the CKF algorithm.
(3) The results show that the algorithm can effectively improve the self-adaptive ability of the integrated system and effectively improve the long-endurance accuracy of the navigation system. The research in this paper has a positive effect on improving the positioning accuracy of the SINS/GPS tightly coupled system and provides a solution to realize the high value of this system for engineering applications.

**Author Contributions:** Conceptualization, J.L. and Y.L.; methodology, J.L.; software, K.D.; validation, K.D., Y.L. and H.P.; formal analysis, H.P.; investigation, J.L.; resources, Y.L.; data curation, K.D.; writing—original draft preparation, J.L.; writing—review and editing, Y.L.; visualization, J.L.; supervision, Y.L.; project administration, H.P.; funding acquisition, Y.L. All authors have read and agreed to the published version of the manuscript.

**Funding:** This research was funded by The Science and Technology Research Program of Chongqing Municipal Education Commission, grant No. KJZD-M202000602.

**Institutional Review Board Statement:** Not applicable.

**Informed Consent Statement:** Not applicable.

**Data Availability Statement:** The experimental site involves some academic project content from our university. For reasons of privacy and academic confidentiality, data sharing is not applicable to this article.

**Conflicts of Interest:** The authors declare no conflict of interest.

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
