# Peer review of "A Tight Coupling Algorithm for Strapdown Inertial Navigation System (SINS)/Global Positioning System (GPS) Adaptive Integrated Navigation Based on Variational Bayesian"

_sustainability, doi:10.3390/su151612477_

Round 1
Reviewer 1 Report
1.This paper introduces a fusion algorithm, which is used to select and fuse the data of SINS and GPS signal sources, and corrects the information obtained from the two systems as the input of the fusion filter, and reduces the influence of noise on the positioning and navigation accuracy in the process. The system is simulated and the noise design is carried out, and the good effect is shown.
2.In the final demonstration of the experimental effect of the combined system, there is a lack of specific description of the experimental environment and rules, such as the motion state of the experimenter and the fixed position of the combined system. In addition, the real-time problem of the combined system is not described in detail.
3.The variable r used in the formula (14) in line 202 is mentioned in the later explanation of its value range, but there is no detailed explanation as to whether the specific value size used in different cases can be explained.
4.In tight coupling mode, the main errors from GNSS are the delay or interference of GPS signals, and the main errors from INS are stochastic sensor biases and drifts. Meanwhile, the existence of ‘time’ difference of the results between GNSS and INS, how to align at their biases in time level. When GPSS signals is lost for long time how to fuse them.
5.The expression of VB-CKF algorithm adaptive fusion flow chart in Fig.2 is unclear.
6.Some other experiments may be needed to be done to compare the advanteges of this paper with other tight coupling algorithms.
The english expression is clear.
Author Response
Dear reviewer, thank you very much for your careful reading of our manuscript and your valuable comments. After getting your suggestions, we have revised and supplemented the contents of the manuscript. In addition, we have also made detailed replies to your doubts in the attachment. Thank you again for your help, and we look forward to your valuable suggestions for our revised manuscript in the future. The following is the attachment we submitted, thank you for taking the time to read.

Reviewer 2 Report
The manuscript titled "A tight coupling algorithm for strapdown inertial navigation system(SINS)/ Global Position System(GPS) adaptive integrated navigation based on variational Bayesian" presents a hybrid positioning method, utilizing variational Bayesian method together with Cubature Kalman Filtering.
The hybrid positioning, combining several sensor information is a relevant research topic currently, when more reliable and accurate positioning methods are all the time needed for self guided vehicles and mobile robots. Enhanced Kalman Filter (EKF) is the traditional method used for sensor fusion in position estimating problems. As stated by the authors, EKF can have problems if the non-linearity is high or when the sensor noise statistics suddenly change. To overcome these issues, the authors propose using Cubature Kalman Filter (CKF) with Variational Bayes (VB) methods. CKF which is known to tolerate higher non-linearity than EKF and VB is used for tracking the changing sensor variance. CKF is used also in previous studies, but the VB-CKF combination is probably novel, and it should make the estimation better adapted to changing noise variance conditions.
The method has been tested with some simulated data and with some real measurements, and compared with the standard CKF. The results show that the proposed VB-CKF is somewhat more accurate than standard CKF
Some comments:
1) Comments on language in the next field.
2) The texts in the figures is in general too small. For example texts in Fig.1, Fig.2, and, Fig.3 is unreadable without heavy zooming. Use approximately the same font size than in the text.
3) There are many relevant prior articles which is not cited in the document, including article, where CKF has been introduced and some other articles which lay the foundation of using CKF for sensor fusion in positioning.
4) Some formulas are seriously wrong. For example equations (1), (2), and (6). Check the others as well.
5) Introduce your variables before showing equations. There are different conventions in the literature and it is not that easy for a reader to understand them if he/she is not certain what the variables are. For example x, \hat{x}, y, f, F, P, Q, h, ...
6) The figure 1, and 2, showing the tight coupling with CKF are not very good. Fig 1 does not show the feedback needed for tight coupling and fig 2 is not complete. For example, the output of CKF prediction is not used at all.
7) The text in results at least in lines 248 - 271 should actually belong to "Materials and methods" since it is describing the testing data and test experiment. The results are in 272 - 286. You can consider or moving some interpretation of the results to the "Discussion", if you like.
8) In KF filtering, the term sigma-point means a few points around the operating point of UKF, to define the distributions. It can be quite misleading that here the term sigma-point is used in a different purpose in line 280 and in figures 3 and 4.
I was not able to go trough formulas 4-24 very carefully. When the formulas which are not correct are fixed and variables explained, then it is easier to do.
The main idea of handling the dynamic noise variation using variational Bayesian technique to improve CKF sensor fusion is interesting, and the manuscript potentially contains some novelty. But unfortunately, in my opinion, it requires a lot of improvements and corrections before it could be considered to be published. But because the idea is acceptable, and the testing environment is good, and the results looks promising, I would like to see this manuscript corrected and published.
More comments are included in the attached PDF document.

I am not a native English speaker, but the language requires more attention. The structure of some sentences is uncomprehensible. For example, sentences starting from line 53 are not understandable:
"so there is no positioning cumulative error caused by the increase of working time. However, due to the need to receive real-time signals during operation, it is easily affected by the external environment.".
Another example, starting from line 64:
"Loose coupling is a simple structure of integrated navigation system, GPS/SINS two systems do not interfere with each other, work independently each other, only GPS navigation data is used to correct SINS system errors."
This is easier to understand, but some verbs or particles are obviously missing.
What does the word "ephesus" mean in line 72?
These needs to be corrected, because they make the text difficult to understand.
Author Response

(The authors gave the same response as above.)

Reviewer 3 Report
The manuscript's objective is the evaluation of the variational Bayesian adaptive estimation method within the cubature Kalman filter with respect to GPS/INS positioning. The content is interesting, but the novelty of the manuscript is not emphasized properly. Poor language has been used, so a thorough proofreading is needed. In addition, the introduction is incomplete, for which specific comment is given below. The numerical results and their analysis should be strengthened, first, in terms of the statistical indices used for the performance evaluation and, second, by using real-world representative data to show the improvement of VB-CKF against other filters. As such, the paper cannot be accepted as is, and a major revision is required that will address the following comments:
1) There is a typo in the title; it should be Global Positioning System (GPS), not Global Position System (GPS).
2) P2, L70: Do the authors mean "pseudorange" instead of "pseudorandom" ? Please clarify.
3) P2, L94: The reference to A. Almagbile is missing. Please correct this.
4) P2, L95: Please define what IMM is.
5) P2, L69 - P3, L101: In this paragraph, the authors describe in detail the various Kalman-filter variants and their (dis)advantages. However, this introduction is incomplete as it lacks information about the Generalized Kalman Filter (https://doi.org/10.1007/s00190-021-01562-0), which allows for a relaxed dynamic model and can provide rigorous solutions when a subset of the state-vector entries are not linked in time.
6) P3, Fig. 1: Is is not clear why only pseudorange measurements are used. Why didn't the authors also opt for using carrier-phase measurements to provide higher accuracy results ? Please elaborate.
7) P4, L155: The authors state that v_k = y_k - h_k (x_k) stands for the accurate measurement noise at the current moment. This is incorrect. The vector v_k stands for the predicted residual vector. Please correct this.
8) P6, L235: I suggest that the authors further elaborate on the selection of the \rho value based on different application environments.
9) P7, L269: How can the measurement noise variance of the pseudorange rate have units m/s? Please check this.
10) P7, L271: Setting the atmospheric delay errors to constant values is not realistic, neither for code-based positioning nor for code-plus-phase-based positioning. Could the authors further explain the impact of this assumption of the user performance ?
11) P8, Fig. 3: Could the authors check whether the legend entries of the figure are correct ? One would expect that the noisy timeseries refers to the estimated values.
12) P9, Fig. 4: The position errors seem to be nonzero-mean for \delta \lambda and \delta H. Could the authors provide the reason behind this ?
13) P10, Tab. 1: The authors should mention what is the statistical index used for the estimation errors shown in Table 1. Do they list the average ? The RMS ? STD ? 95% percentile ? Please specify.
Due to the poor English language used, extensive proofreading is required.
Author Response

(The authors gave the same response as above.)

Round 2
Reviewer 3 Report
I would like to thank the authors for improving their manuscript. Unfortunately, not all of my remarks have been seriously addressed. The authors are asked to elaborate again on the following remarks:
- S5
- S6: the authors present some reasons/excuses of not using phase data, which are not accepted, since they do not pose as blocking points to the conducted research. Please make use of both code and phase data for your analysis.
- S8: it is still not clear how the users can a priori select the \rho value, depending on their case, measurement-quality, environment, etc. Please provide a reasoning behind the selection of this value.
- S10: the fact that the atmospheric delay errors are not accurately modeled and thus set to constant diverges significantly from reality, and certainly leads to results that are a) not reliable, b) not representative of the real-world conditions. Please take proper care in modeling such delays to obtain a realistic simulation analysis, and re-compute your solutions
- English: Still, the text is not coherent. Please revise it.
Author Response

(The authors gave the same response as above.)

Round 3
Reviewer 3 Report
The paper can now be accepted for publication.
Some improvements in coherency are still needed.